# Genome analyses reveal population structure and a purple stigma color gene candidate in finger millet

Katrien M. Devos [1,2,3,29] ✉, Peng Qi[1,2,3,29], Bochra A. Bahri[1,4], Davis M. Gimode[1,5], Katharine Jenike[6], Samuel J. Manthi[5,22], Dagnachew Lule[2,7,23], Thomas Lux[8], Liliam Martinez-Bello[1,2,3,24], Thomas H. Pendergast IV [1,2,3], Chris Plott [9], Dipnarayan Saha [2,25], Gurjot S. Sidhu[1,2,3], Avinash Sreedasyam [9], Xuewen Wang[10], Hao Wang[10], Hallie Wright[1], Jianxin Zhao[1,2,3], Santosh Deshpande[11,26], Santie de Villiers[12,13], Mathews M. Dida[14], Jane Grimwood [9], Jerry Jenkins [9], John Lovell [9,15], Klaus F. X. Mayer [8,16], Emmarold E. Mneney[17,27], Henry F. Ojulong[18], Michael C. Schatz [6], Jeremy Schmutz [9,15], Bo Song [19,28], Kassahun Tesfaye [20,21] & Damaris A. Odeny [5]

Finger millet is a key food security crop widely grown in eastern Africa, India and Nepal. Long considered a 'poor man's crop', finger millet has regained attention over the past decade for its climate resilience and the nutritional qualities of its grain. To bring finger millet breeding into the 21st century, here we present the assembly and annotation of a chromosome-scale reference genome. We show that this ~1.3 million years old allotetraploid has a high level of homoeologous gene retention and lacks subgenome dominance. Population structure is mainly driven by the differential presence of large wild segments in the pericentromeric regions of several chromosomes. Trait mapping, followed by variant analysis of gene candidates, reveals that loss of purple coloration of anthers and stigma is associated with loss-of-function mutations in the finger millet orthologs of the maize *R1/B1* and *Arabidopsis GL3/EGL3* anthocyanin regulatory genes. Proanthocyanidin production in seed is not affected by these gene knockouts.

Finger millet, *Eleusine coracana* (L.) Gaertn subsp. *coracana*, is an important subsistence crop for smallholder farmers in eastern Africa and India. For more than a century, global political influences and government policies favored the production of maize in eastern African agricultural systems, relegating finger millet to the status of 'poor man's crop'. Because finger millet was grown during most of the 20th century as an insurance crop rather than the main staple in eastern Africa, little attention was given to genetic improvement. Consequently, yields were typically low owing to the use of unimproved landraces and traditional farming practices. Furthermore, breeding of

finger millet, an inbreeding species that is difficult to cross, has traditionally been done by selecting lines with improved performance[1], resulting, presumably, from rare spontaneous outcrossing events. Hybridization-based breeding, which started in India in the 1950s, has been practiced routinely in eastern Africa only since the early 2000s by a small number of breeders. Breeding remains hampered, however, by the limited information available on the genetic diversity of finger millet germplasm, a dearth of marker-trait associations and, more generally, a lack of genetic and genomic resources[2]. These resources are needed to efficiently identify the chromosomal regions, genes and

pathways that play key roles in target traits such as biotic and abiotic stress tolerance, grain yield and grain nutritional value.

Here, we report on the generation of a high-quality genome assembly of allotetraploid finger millet, the population structure of a finger millet germplasm collection from across the finger millet growing regions of Africa and South Asia, and the application of the sequence information to identify the causal gene candidate for a quantitative trait locus (QTL) for anthocyanin production in stigma and anthers.

## Results

### The allotetraploid finger millet genome

Finger millet is a primarily inbreeding allotetraploid species (AABB genome; $2n = 4x = 36$) belonging to the family Poaceae, subfamily Chloridoideae. Two existing short read-based genome assemblies for Indian cultivars ML-365[3] and PR202[4] have proven useful for broad-scale analyses. However, finger millet and Chloridoideae in general would benefit from a more complete and contiguous genome resource, especially one that represents the germplasm in eastern Africa where finger millet was domesticated. To accelerate molecular breeding and trait discovery, we generated a reference-quality chromosome-resolved assembly for the cultivated Kenyan finger millet accession KNE 796 combining long and short-read sequencing. In short, an initial assembly from 152.47 Gb of PacBio long-read sequences (84.71x genome coverage; 6981 bp average read length) was scaffolded with 164.2x coverage Hi-C, and polished with 94.6x Illumina 2 × 150 bp reads. A 4400-marker genetic map was used to validate and identify additional contig joins (see methods for assembly details). With 18 chromosomes and 1.11 billion bases (Gb), the assembled size of the KNE 796 v1.0 genome is similar to previous assemblies but far more contiguous (contig N50 = 15,268 kb vs 285 kb[4] and 24 kb[3]; Supplementary Table 1). The quality value (QV) score for the assembly is 58.8 (Supplementary Note 1). The predominantly selfing nature of finger millet is reflected in the low heterozygosity observed in the genome assembly (1 heterozygous single nucleotide polymorphism (SNP) every 12.3 kb). This extremely low level of heterozygosity means that KNE 796-S (the -S is added to avoid confusion with a similarly named but genetically distinct KNE 796 accession that is in circulation) is best represented by a haploid-collapsed and not by a haplotype-resolved reference genome. Complete genome-level statistics are given in Supplementary Table 2 and the v1.0 genome is available from Phytozome (https://phytozome-next.jgi.doe.gov/info/Ecoracana_v1_1).

Alignment of resequencing data from two accessions of *Eleusine indica*, the diploid A-genome donor to finger millet[5,6], demonstrated that seven chromosomes had an A-genome origin and two carried a reciprocal homoeologous translocation (Supplementary Fig. 1; Supplementary Note 2). Bar a number of small inversions, and one large inversion that differentiates chromosomes 3A and 3B, the homoeologous A and B chromosomes are highly colinear (Supplementary Fig. 2). All finger millet chromosomes are metacentric, except chromosomes 4A and 4B which are acrocentric (Supplementary Fig. 3), a conformation that likely dates back to the grass ancestor[7].

The B-genome chromosomes are, on average, 20% longer than the A-genome chromosomes (Supplementary Fig. 2), caused by a higher content of repetitive DNA (Supplementary Table 3; Supplementary Note 3). The overall repeat content of the finger millet genome is 61.3%, with 74% of the repeats being long terminal repeat retrotransposons (LTR-RTs). The prevalence of different repeat classes in the finger millet genome is shown in Table 1. Full-length elements in eight and 19 LTR-RT families (number of full-length elements per family ≥ 5) are uniquely present in the A genome (total of 78 intact elements covering 0.7 Mb) and B genome (total of 896 intact elements covering 7.5 Mb), respectively (Supplementary Table 4). An additional 11 families have significantly ($p < 0.05$) more intact elements in one compared to the other subgenome with four families being overrepresented in the A

genome and seven in the B genome. The total number of intact LTR-RTs across the 11 families is, however, similar in both subgenomes (1463 in the A genome *versus* 1481 in the B genome) (Supplementary Table 5).

Gene numbers, that is, 48,836 high confidence (HC) genes equally distributed across the A and B genomes (24,287 and 24,310 genes, respectively), and 24,176 low confidence (LC) genes (47.9% on the A genome and 52.1% on the B genome) (see Supplementary Table 6 for detailed annotation statistics), are in the expected range for an allotetraploid species. Completeness of the annotation was provided by an overall Benchmarking Universal Single-Copy Orthologs (BUSCO)[8] score of 97% with 95.2% for the HC genes and 7.3% for the LC genes (Supplementary Fig. 4). The BUSCO values suggest that some bona fide genes were classified as LC genes, which is also indicated by the fact that some 2% of LC genes displayed ≥90% similarity to and covered ≥90% of the homoeologous HC protein. Of the 97% of complete BUSCO genes identified in the finger millet genome, 83.1% were duplicated and 13.9% were single copy. The higher percentage of genes present on both subgenomes in the KNE 796-S assembly compared to the previously published PR202[4] assembly (83.1% vs. 56.3%; Supplementary Table 7) is a further indication of the high quality of our assembly.

### Diversification of finger millet in distinct germplasm pools

Finger millet is reported to have been domesticated some 5000 years ago (YA) from the wild *Eleusine coracana* subsp. *africana* in the highlands that stretch from Ethiopia to Uganda[9,10], from where the crop subsequently dispersed to the African lowlands and then to India. Archaeobotanic evidence dates the presence of finger millet in India to around 2000–2500 YA[11], and it is generally accepted that the South Asian germplasm pool remained largely separate from the African germplasm pool until the 1950s when intercrossing between the two pools led to the popular Indaf varieties[12]. We conducted a population structure analysis on 308 *E. coracana* subsp. *coracana* and 22 *E. coracana* subsp. *africana* accessions from the main finger millet growing regions in Africa and South Asia using SNPs mined from genotyping-by-sequencing (GBS) data or using the corresponding SNPs from whole genome resequencing data. The SNPs were filtered to have a read depth per sample ≥8x and were distributed across the genome (Fig. 1; Supplementary Data 1-2). The optimal number of subpopulations was two (k = 2), regardless of whether we used A-genome SNPs ($n = 6185$) or B-genome SNPs ($n = 4592$), and separated wild (subsp. *africana*) from cultivated (subsp. *coracana*) accessions (Supplementary Fig. 5). The five *E. indica* (diploid, A-genome relatives) accessions that we included in the A-genome analysis grouped with the subsp. *africana* accessions. Accessions with >75% membership to the wild subpopulation in both the A- and B-genome analyses were considered true wild accessions (referred to as pop0-pop0). Surprisingly, accessions originating from Asia were admixed (38% wild/62% cultivated) between the wild and cultivated germplasm pools in the A-genome but not the B-genome analysis (membership wild/cultivated of 1%/99%) (Supplementary Fig. 5). The mixed membership was caused by the presence of large pericentromeric segments carrying wild alleles, most notably on chromosome 5A (Fig. 2). Repeating the population structure analysis on accessions with ≤75% membership to the wild subpopulation identified two subpopulations (k = 2) with the A-genome SNPs and four subpopulations (k = 4) with the B-genome SNPs (Supplementary Fig. 6). Subpopulation membership was geographically structured with the A-genome SNPs differentiating Asian from African germplasm (Supplementary Fig. 6A). The B-genome SNPs largely clustered accessions into an Ethiopian group, two African groups, and a mixed African and Asian group with no obvious geographic stratification of the non-Ethiopian African germplasm (Supplementary Fig. 6B). Combining the information across the two genomes and excluding admixed lines (≤75% membership to a single

**Table 1 | Summary of different repeat types and their prevalence**

| Type | Subclass | Superfamily | No. of families | No. of elements | Length (Mb) | % of genome |
|---|---|---|---|---|---|---|
| Class I: Retroelements | LTR-RT | Gypsy | 1345 | 277,258 | 349.9 | 31.5 |
| | | Copia | 422 | 117,165 | 153.7 | 13.8 |
| | | Unknown | 161 | 78,351 | 80 | 7.2 |
| | LINE | LINE | 4655 | 92,325 | 26.3 | 2.4 |
| | SINE | SINE | 41 | 7378 | 1.3 | 0.1 |
| Class II: DNA Transposons | TIR | Mutator | 35 | 5549 | 1 | 0.1 |
| | | PIF | 72 | 8778 | 1.8 | 0.2 |
| | | hAT | 89 | 13,004 | 2.6 | 0.2 |
| | | CACTA | 215 | 26,852 | 4.7 | 0.4 |
| | | Tc1 | 959 | 112,087 | 20.8 | 1.9 |
| | | Other | 1 | 143,640 | 6 | 0.5 |
| | Helitron | – | 362 | 123,538 | 27.4 | 2.5 |
| | Tandem repeats | – | 1 | 143,640 | 6 | 0.5 |
| | Total | – | | | 681.5 | 61.3 |

subpopulation), resulted in one wild and five cultivated germplasm groups referred to as pop0-pop0 (wild), pop1-pop1 (Ethiopian), pop1-pop2 (mixed African 1), pop1-pop3 (mixed African 2), pop1-pop4 (mixed African 3) and pop2-pop2 (Asian) (Supplementary Data 3; Supplementary Fig. 7).

A comparison of the alleles present in cultivated and wild sub-populations for ~10,700 GBS-SNPs (206 accessions) and ~3.5 million resequencing SNPs (27 accessions) indicated that subpopulation formation was largely driven by the presence of large pericentromeric clusters of alternate (non-reference) alleles on chromosomes 4A, 5A, 7A, 8B and 9B (Fig. 2; Supplementary Fig. 8; Supplementary Data 4). This is supported by the varying population membership at k = 2 when analyses are done by chromosome for accessions with >75% membership to the six population groups, as observed for, for example, pop2-pop2 (Asian) for chromosome 5A, pop1-pop4 (mixed African 3) for 8B and pop1-pop3 (mixed African 2) for 9B (Supplementary Fig. 9). Wild accessions appear as admixed in this analysis for chromosomes 4A and 7A. While cultivated accessions carry the reference allele at most loci and wild accessions the alternate allele, this is not the case for the pericentromeric regions of chromosomes 4A and 7A (Fig. 2; Supplementary Fig. 9). Plotting the relative allele frequencies in the two resequenced *E. indica* accessions showed that both carried the alternate allele on chromosome 4A and the reference allele on chromosome 7A (Supplementary Fig. 10). The same allele configuration was found in the two resequenced Ethiopian wild lines, AAUELU-13 and AAUELU-46, while the reverse configuration was found in MD-20, a wild Kenyan accession (Supplementary Fig. 10). Other wild lines carried a mix of reference and alternate alleles in those regions. We hypothesize that the alleles present in *E. indica* and the two Ethiopian accessions represent the ancestral haplotype. Reinterpreting the SNP results based on this assumption (Fig. 2; Supplementary Fig. 8), we conclude that retained wild pericentromeric segments are present on chromosome 4A predominantly in Ethiopian (pop1-pop1) and Asian accessions (pop2-pop2), and on chromosome 7A predominantly in African accessions (pop1-pop1, pop1-pop2, pop1-pop3 and pop1-pop4) (Supplementary Table 8; Supplementary Fig. 9). Given the phylogenetic scale of our sequencing, it is not possible to discern whether some of the wild segments date back to the domestication of finger millet or are the result of subsequent intercrossing between the two subspecies which grow in sympatry in eastern Africa. Retention of large pericentromeric regions over several thousand years is conceivable, considering the limited opportunity for recombination caused by the inbreeding nature of finger millet, the selection-based breeding approach that is common in eastern Africa and the suppression of recombination in pericentromeric regions (Supplementary

Fig. 3). In Asian accessions (pop2-pop2), any wild segments would have had to be present at the time the lines were introduced into South Asia because the wild subsp. *africana* is limited in its distribution to Africa. Their retention across Asian germplasm suggests that the germplasm pool that was brought from Africa to South Asia was small with limited diversity.

The wild pop0-pop0 population had similar levels of differentiation from all five cultivated subpopulations (Supplementary Table 9). This was also generally observed when conducting the divergence analysis on a chromosome-by-chromosome basis, except for chromosomes that carried large wild segments (Supplementary Data 5-8). This was surprising because Asian germplasm (pop2-pop2) has remained largely isolated from wild germplasm, which only occurs in Africa, since its introduction in India some 2500 YA[13]. The divergence results suggest that gene flow in Africa is low between wild and cultivated lines. Asian finger millet germplasm (pop2-pop2) had the lowest Fst value at the population level with pop1-pop2, concomitant with a lowland African origin. The overall lowest genetic differentiation (Fst and Dxy) was observed between the African subpopulations pop1-pop2 and pop1-pop4, which cover largely the same geographic area.

**Retracing the birth of finger millet**

The Chloridoideae subfamily comprises as many as 166 genera and more than 1500 species[14]. Superimposing information gained from comparative analyses of the KNE 796-S v1.0 assembly with those of the Chloridoideae species *Oropetium thomaeum* ($2n = 20$)[15] and *Eragrostis tef* (teff; $2n = 4x = 40$)[16], the Panicoideae species *Sorghum bicolor* (sorghum; $2n = 20$)[17] and the Ehrhartoideae species *Oryza sativa* (rice; $2n = 24$)[18] on known grass genome relationships[19,20] (Fig. 3) showed that the divergence of the Chloridoid lineage was accompanied or driven by a reduction in chromosome number from the presumed grass ancestral number of 12[21] to 10 through insertional dysploidy, the most common form of chromosome reduction in the grasses[22,23] (Supplementary Table 10). Fusions involved ancestral chromosomes 9 and 10, the same chromosomes that participated in the 12 to 10 reduction in chromosome number that led to the Panicoid lineage[19], although the acceptor chromosomes were different (Supplementary Table 10). Chromosomes 9 and 10 are the smallest chromosomes in rice, which is used as the ancestral model[21], and we hypothesize that insertional dysploidy may occur more readily with smaller chromosomes.

A further chromosome reduction, again through insertional dysploidy, from $x = 10$ to $x = 9$, gave rise to the genus *Eleusine* in the subtribe Eleusinineae[24]. The fusion involved what is currently teff chromosome 10 (corresponding to rice chromosome 12) and chromosome 9 (syntenic to rice chromosome 5) to yield finger millet

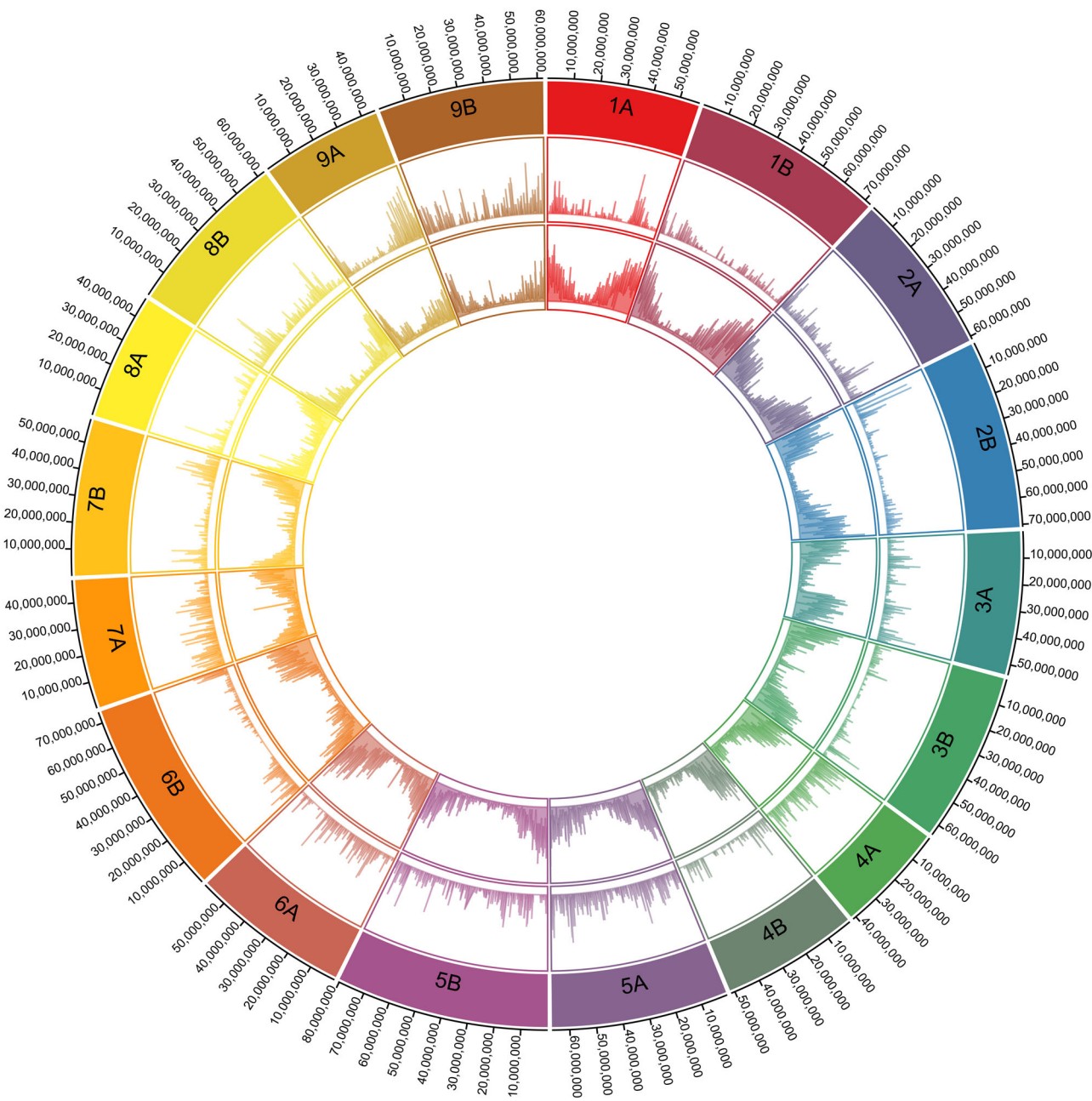

**Fig. 1 | Circos diagram showing the distribution of SNPs used in the population structure analysis across the finger millet genome.** The rings, from outside to inside, represent the 9 A-genome and 9 B-genome chromosomes (each in a different color), the number of SNPs per 100 kb (min = 0, max = 25) and the number of genes per 100 kb (min = 0, max = 25). Source data are provided as a Source Data file.

chromosome 5 (Fig. 3; Supplementary Table 10). Interestingly, this chromosome reduction in *Eleusine* involved the same two ancestral chromosomes that were fused to reduce the chromosome number in the Panicoid tribe *Paniceae* from 10 to 9[25], although the configuration was 5 – 12 – 5 in finger millet and 12 – 5 – 12 in the *Paniceae* species switchgrass using rice chromosome nomenclatures as the ancestral designations (Fig. 3).

The genus *Eleusine* comprises nine known species, eight of which are native to Africa, including *Eleusine coracana* and *E. indica*[26]. The wild progenitor of finger millet, *E. coracana* subsp. *africana*, originated through the hybridization of *E. indica* (AA genome) with an unknown and possibly extinct B-genome donor[5,6]. Using the number of synonymous substitutions between 16,448 homoeologous finger millet genes with a 1:1 relationship (Supplementary Data 9) and the grass

synonymous substitution rate of 6.5×10⁻⁹ substitutions per synonymous site per year[27], the divergence between the A and B genomes, which likely reflects the divergence between the A- and B-genome progenitors, was estimated at 6.25 million years ago (MYA) (Supplementary Fig. 11). The timing of the tetraploidization event can be gleaned from the insertion dates of genome-specific LTR-RT families. As expected, very young elements with identical LTR sequences were found only in retrotransposon families with intact elements in both subgenomes (Supplementary Fig. 12). The peak of transposition activity of the three B-genome-specific families that comprised more than 80 elements was around 2 MYA with the 25% youngest transposition events dating back, on average, 1.3 MYA. We used the latter number as the presumed date of the tetraploidization event. The young age of finger millet concurs with the high level of retention of

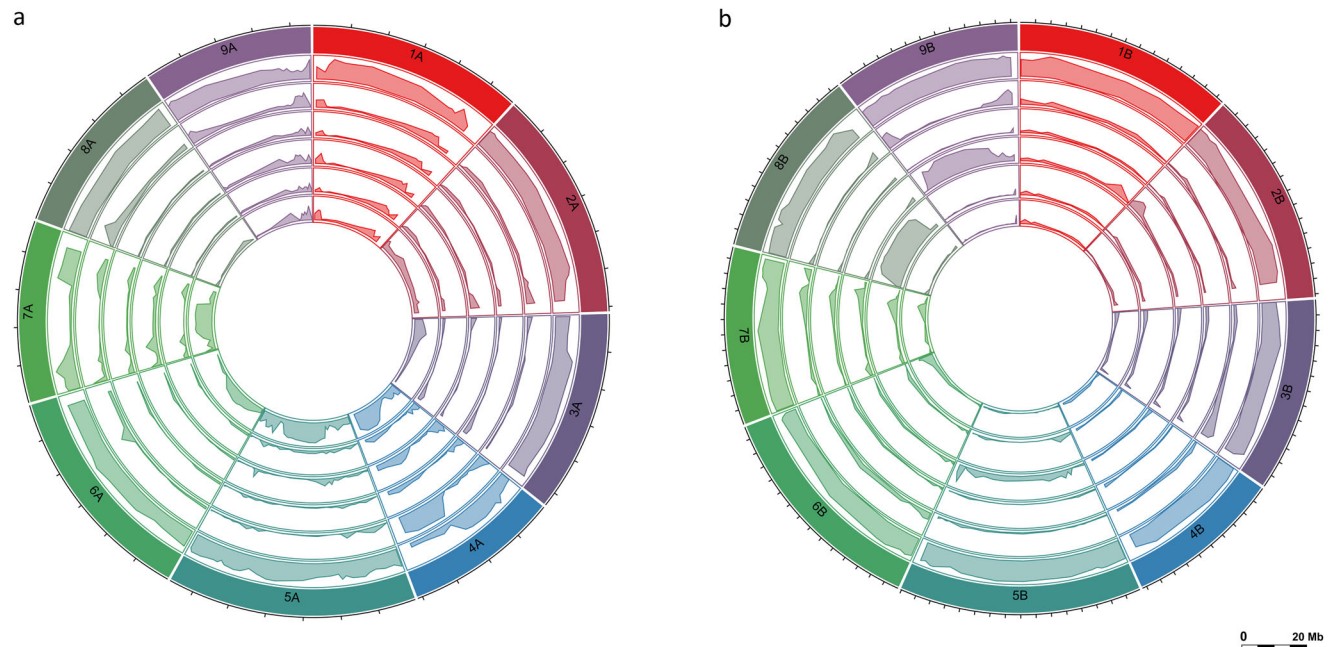

**Fig. 2 | Circos diagrams showing the presence of wild alleles.** The diagrams show the percentage of alternate alleles (minimum: 0%; maximum: 100%) relative to the KNE 796-S reference genome per 50 SNP window along each chromosome (outer ring) in (**a**) the A genome and (**b**) the B genome in the six identified finger millet subpopulations. Subpopulations, from 2nd outer ring to inner ring are pop0-pop0 (wild), pop1-pop1 (Ethiopian), pop1-pop2 (mixed African 1), pop1-pop3 (mixed African 2), pop1-pop4 (mixed African 3) and pop2-pop2 (Asian). Each chromosome is depicted in a different color. As indicated by the scale bar, the distance between tick marks on the outer (chromosome) ring represents 5 Mb. The accessions comprised within each subpopulation can be found in Supplementary Data 3. Source data are provided as a Source Data file.

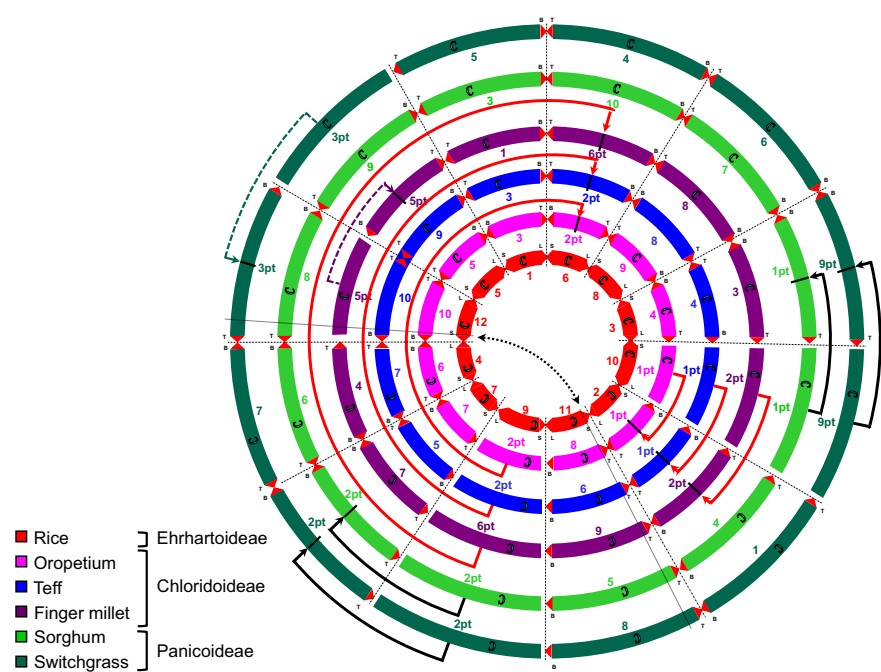

**Fig. 3 | Chromosome-level relationships of select grass species.** The diagram shows the relationships of the chromosomes of the Chloridoideae species *Oropetium thomaeum*, *Eragrostis tef* and *Eleusine coracana*, and the Panicoideae species *Sorghum bicolor* and *Panicum virgatum* relative to the Ehrhartoideae species *Oryza sativa*. Chromosome fusions that occurred in the Chloridoid ancestor are shown with red arrows and in the Panicoid ancestor with black arrows. Chromosome fusions indicated in dotted lines are species-specific. Source data are provided as a Source Data file.

homoeologous gene copies. Only 8.9% of single-copy genes, all of which were confirmed to have an ortholog in either rice or sorghum, were uniquely present in either the A genome (5.3%) (Supplementary Data 10) or the B genome (3.6%) (Supplementary Data 11). No subgenome dominance was observed when comparing transcript levels of

homoeologous gene pairs across different tissues (MANOVA, $p = 0.999$; Supplementary Table 11; Supplementary Data 12).

Comparative analyses with *E. indica* revealed that at least two reciprocal translocations occurred between homoeologous chromosomes in tetraploid finger millet, with the 6A/6B

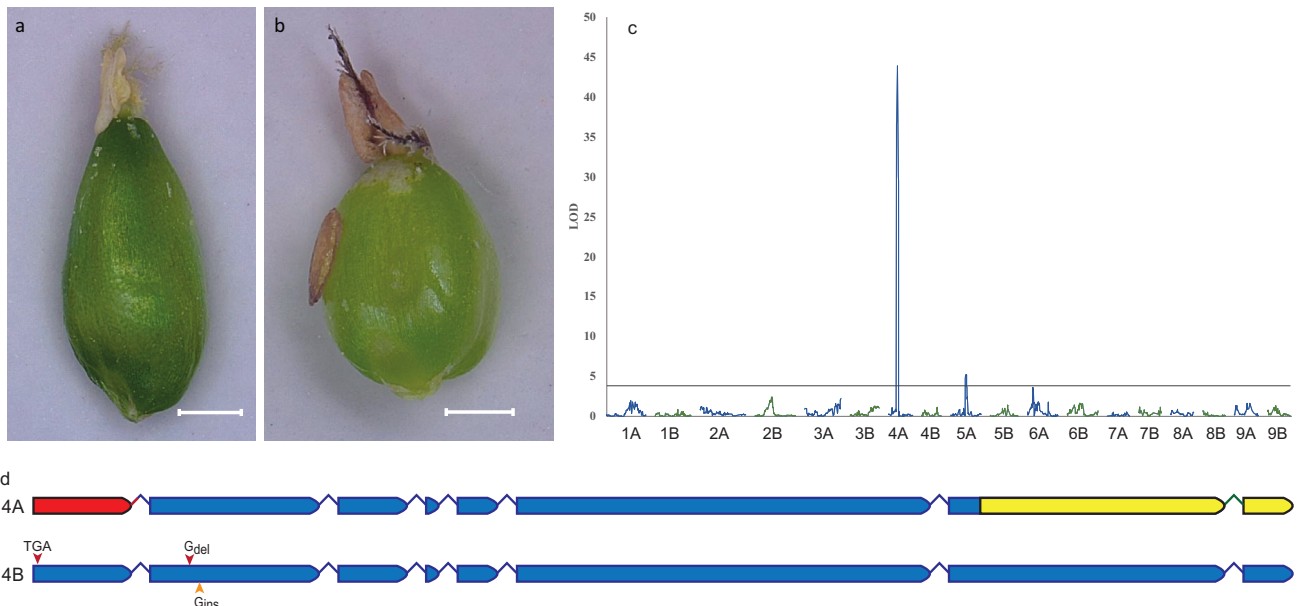

**Fig. 4 | Phenotype, QTL and variants in a candidate gene for purple anthers and stigma. a, b** Microscopic image of (**a**) MD-20 ovule with white stigma and yellow anthers and (**b**) Okhale-1 ovule with purple stigma and anthers. The scale bar is 500 μm. **c** Graph of LOD scores showing the presence of a large-effect QTL for anther/stigma color on chromosome 4A and a small-effect QTL on chromosome 5A in a wild (MD-20) x cultivated (Okhale-1) biparental F₂ population. A-genome chromosomes are shown in blue; B-genome chromosomes are shown in green. The horizontal dashed line represents the LOD significance threshold at α = 0.05 determined by 1000 permutations (LOD = 4.3). The genetic map (Supplementary

Data 13) was updated from Qi et al.[28] using SNPs called against KNE 796-S assembly v1.0. Phenotypes are available from Supplementary Data 13. **d** Schematic representation of non-functional alleles of the *PP* gene on chromosome 4A and its homoeolog on 4B. Introns are not drawn to scale. The 5' deletion in the KNE 796-S reference genome is indicated in red. The 1127 bp 3' deletion in MD-20 is indicated in yellow. Red arrows indicate function-inactivating SNPs in KNE 796-S and Okhale-1, and the orange arrow indicates a function-inactivating SNP in MD-20. TGA = C to T SNP giving rise to a stop codon; Gdel = frameshift mutation caused by a single base deletion; Gins = frameshift mutation caused by a single base insertion.

translation involving ~20% of the chromosome lengths and the 9A/9B translocation ~6%. An interstitial reciprocal 2A/2B translocation, previously identified as differentiating MD-20 and Okhale-1[28] (Supplementary Fig. 3) was not observed in the KNE 796-S genome assembly. The 6A/6B translocation predates finger millet domestication[28] and may be a legacy of the genome instability that sometimes characterizes young allopolyploids[29]. The 9A/9B translocation was absent in wild lines, consistently present in cultivated accessions from countries other than Ethiopia, and showed a mixed presence in Ethiopian cultivars (Supplementary Fig. 7; Supplementary Data 3). The mixed presence indicates either standing variation for the 9A/9B translocation in the cultivated germplasm pool in Ethiopia, the presumed center of domestication[9], at the time when domesticated finger millet dispersed to lowland Africa or more recent loss due to outcrossing with the sympatric subsp. *africana*. Either scenario suggests an early post-domestication origin of the 9A/9B translocation. The breakpoint of the 9A/9B translocation was delineated to a highly conserved 46 bp region located within an intron of the homoeologous genes ELECO.r07.9AG0673650 and ELE-CO.r07.9BG0697350, which were designated low confidence (LC) annotations, but appear to be bona fide genes. While there was considerable sequence divergence in the intron 3' of the breakpoint, the high sequence homology seen in the breakpoint region extended 5' into the rest of the intron and the upstream exon. This concurs with the observation that homoeologous exchange requires high sequence homology and, consequently, typically takes place within genes, particularly in coding regions[30]. The homoeologous translocations suggest that, despite the largely bivalent formation during meiosis[6], pairing control in allopolyploid finger millet may once have been (and may still be) incomplete, leading to chromatid exchanges (crossovers) between homoeologs. The differential presence of the 9A/9B

translocation in wild and most cultivated germplasm, potentially leading to genotypes with unbalanced chromosomes upon intercrossing, may also have contributed to the low level of gene flow observed between subsp. *africana* and *coracana*.

## The genetics of purple coloration of stigma and anthers
The purple coloration of internodes, stigma and anthers is widely used by breeders as a visual marker to identify true F₁ hybrids in crosses between parents that vary in these traits. We investigated anthocyanin production in stigma and anthers as a case study to demonstrate the use of the KNE 796-S genome assembly for the identification of causal genes for traits of interest. Purple plant pigmentation, which is dominant to green, had been attributed to a single factor, PP, in finger millet, although additional factors were predicted to modulate color intensity[31]. QTL analysis of stigma and anther color, which cosegregated in 122 F₂ progeny from the cross MD-20 (subsp. *africana*; white stigma and yellow anthers; Fig. 4a) x Okhale-1 (subsp. *coracana*; purple stigma and anthers; Fig. 4b), identified a large-effect QTL (LOD = 43.9) on chromosome 4A that explained 77.0% of the variation and a small-effect QTL (LOD = 5.2) on chromosome 5A that explained 5.3% of the variation (Fig. 4c; Supplementary Note 4). The Okhale-1 allele at the 4A locus had a positive effect on anthocyanin production, while the allele at the 5A locus had an inhibitory effect. The 5A QTL interval contains two R2R3-MYB transcription factors (ELE-CO.r07.5AG0383440 and ELECO.r07.5AG0383450) with homology to *Anthirrhinum majus* MYB308, a repressor of the phenylpropanoid pathway that may also repress flavonoid biosynthesis[32]. ELE-CO.r07.5AG0383450, in particular, warrants further investigation because MD-20 contains a non-synonymous SNP resulting in an arginine (R) to cysteine (C) substitution in the highly conserved C1 motif[33].

Within the 4A QTL interval, ELECO.r07.4AG0307750, a MYC-bHLH transcription factor orthologous to the maize anthocyanin regulatory genes *R*-S (*Seed color component at R1*; Genbank acc.

P13027.1)[34] and *Lc* (*Leaf color*; Genbank acc. P13526.1)[35] was the only gene within the ~1.6 Mb interval with a gene ontology (GO)-term or description associated with anthocyanin biosynthesis. ELECO.r07.4AG0307750 was located approximately 268 kb from the peak of the QTL. Alignment with maize *R*-S and *Lc*, which are two alleles of the same gene, showed that the protein encoded by ELECO.r07.4AG0307750 is lacking the N-terminal 80 amino acids (Fig. 4d; Supplementary Fig. 13). Alignment of maize *Lc* with the corresponding genomic region of finger millet chromosome 4A that encompassed ELECO.r07.4AG0307750 showed that the region corresponding to exon 1 is missing from the finger millet genome assembly. The deletion was confirmed by amplicon sequencing, and shown to include 110 bp of the presumed 5' UTR, exon 1 (149 bp) and 203 bp of intron 1. Because the deletion included the actual start codon, a downstream ATG located in exon 2 had been annotated as the starting codon in the KNE 796-S genome assembly. The orthologous gene from *E. indica*, which almost universally displays the anthocyanin phenotype in reproductive organs, was extracted from the HZ-2018 genome assembly[36] (Genbank Acc. QEPD01000187) and was found to encode a full-length protein (Supplementary Fig. 13). The Okhale-1 allele also encodes a full-length protein, while the MD-20 allele is truncated at the C-terminus (Fig. 4d; Supplementary Fig. 13). Functional inactivation caused by the observed deletions in ELECO.r07.4AG0307750, which we will refer to as *PP*, in KNE 796-S and MD-20 agrees with those two accessions having lost anthocyanin production in reproductive tissues.

Because purple coloration is dominant and no QTL was identified at the homoeologous locus on chromosome 4B, the expectation was for ELECO.r07.4BG0338780, the 4B homoeolog of ELECO.r07.4AG307750, to be non-functional in both MD-20 and Okhale-1. Similar to its 4A homoeolog, a downstream ATG codon in ELECO.r07.4BG0338780 had been annotated as start codon, thereby computationally masking the presence of upstream function-inactivating mutations. When considering the start codon defined by homology with orthologous maize and *Setaria* sequences, ELECO.r07.4BG0338780 in KNE 796-S contained a C to T SNP relative to the wild accessions EA_Serere and MD-20 early in exon 1, which resulted in a stop codon, and a single base (G) deletion in exon 2 which led to a frameshift mutation (Fig. 4d). No transcripts were identified for ELECO.r07.4BG0338780 in either RNASeq data generated from different tissues of KNE 796 (SRR5341138 – SRR5341148) or by reverse transcription polymerase chain reaction (RT-PCR) using gene-specific primers, presumably because non-functional transcripts undergo rapid degradation through the nonsense-mediated mRNA decay machinery[37]. The same two mutations, which were confirmed by amplicon sequencing, were observed in other resequenced cultivated accessions, including Okhale-1. Interestingly, wild accessions, including MD-20, carry a 1-bp insertion (G) 35-bp downstream of the 1-bp deletion present in cultivated germplasm. Non-functionality of ELECO.r07.4BG0338780 in both MD-20 and Okhale-1 explains the lack of anthocyanin in anthers and stigma in MD-20 (which also carries a non-functional ELECO.r07.4AG307750 allele), as well as the lack of a QTL on chromosome 4B in the MD-20 x Okhale-1 population. The presence of the 5' or 3' deletion in the *PP* coding region was assessed by PCR in a subset of the finger millet germplasm and correlates perfectly with the presence of white stigma and yellow anthers ($n = 15$). This also demonstrates that nonfunctionality of the 4B homoeolog is widespread in finger millet germplasm. Conversely, 15 out of 22 accessions (68%) with a presumed full-length *PP* gene displayed purple coloration. The remaining lines likely carry other non-functional alleles of *PP* or deleterious mutations in other genes of the anthocyanin biosynthesis pathway. Indeed, variant mining of known anthocyanin regulatory and biosynthetic genes in whole-genome shotgun sequence data generated for IE2244, a line with white stigma, yellow anthers and white seed and carrying a full-length copy of the *PP* gene, revealed function-inactivating mutations in exon 2 of the homoeologous chalcone synthase genes ELECO.r07.9AG0686670 (nonsense mutation) and ELECO.r07.9BG0712910 (1-bp insertion). Both mutations were validated by PCR (Supplementary Fig. 14). The same two mutations were present in White Sel6, another white-seeded line with white anthers and yellow stigma and lacking the *PP* partial deletion, suggesting that inactivation of the homoeologous chalcone synthase genes on the group 9 chromosomes abolishes anthocyanin biosynthesis in both reproductive organs and the seed testa. Collectively, our results strongly support our hypothesis that *PP* (ELECO.r07.4AG0307750) controls anthocyanin production in stigma and anthers in finger millet.

We further assessed whether *PP* allele status correlates with flavonoids in the seed testa, which is typically pigmented in finger millet and also contains high levels of proanthocyanidins (condensed tannins)[38]. Proanthocyanidins are oligo- or polymers of (epi)catechin units derived from leucoanthocyanidin[39], which forms the branchpoint between the anthocyanin and proanthocyanidin pathways[40]. If the anthocyanin/proanthocyanidin pathways in seed and reproductive organs are controlled by a common bHLH transcription factor, finger millet accessions that lack a functional *PP* gene and, consequently, are devoid of anthocyanins in stigma and anthers would be expected to also lack seed proanthocyanidins. This has implications for breeding for seed quality because proanthocyanidins have been associated with multiple health benefits[41]. Quantification of condensed tannins using the vanillin assay in ground seeds of MD-20, Okhale-1 and select progeny that were either homozygous for a functional *PP* allele (purple stigma and anthers) or homozygous for a truncated allele (white stigma/yellow anthers) showed that the presence of proanthocyanidins in the seed coat was independent of the allele status of *PP* (Supplementary Table 12). Anthocyanin production in stigma and anthers, and proanthocyanidin biosynthesis in the grain in finger millet are thus controlled by different bHLH transcription factors.

A maximum likelihood tree of the closest homologs in finger millet (KNE 796-S v1.0), maize (RefGene_V4)[42], foxtail millet (v2.2)[43] and rice (Kitaake v3.1)[44] to *Arabidopsis* proteins GLABRA 3 (GL3), ENHANCER of GLABRA 3 (EGL3) and TRANSPARENT TESTA 8 (TT8), which have partially redundant functions in anthocyanin and proanthocyanidin biosynthesis[45], showed that finger millet PP, maize homoeologs r1 (Red1; Zm00001d026147; allelic to *R*-S and *Lc*) and b1 (Booster 1; Zm00001d000236) clustered with *Arabidopsis* GL3 and EGL3 (Supplementary Fig. 15). A sister clade comprised *Arabidopsis* TT8, finger millet homoeologs ELECO.r07.7AG0560560 and ELECO.r07.7BG0592150 (annotated incorrectly structurally and as a low confidence gene in assembly v1.0), and the rice protein OsKitaake07g072000 (Rc) which controls proanthocyanidin accumulation in the pericarp of brown-colored rice grains[46]. It should be noted that finger millet grain is an utricle with a membranous pericarp, and that the pigmentation and proanthocyanidins are located in the testa. Also included in the TT8 clade was maize INTENSIFIER1 (IN1; Zm00001d019170). IN1, when inactivated, increases flavonoid production in the aleurone and may act as a competitive inhibitor for R1[47]. We hypothesize that, despite the varying levels of subfunctionalization and, potentially, neofunctionalization of the orthologs in the TT8 clade, ELECO.r07.7AG0560560 and ELECO.r07.7BG0592150 control the production of anthocyanins/proanthocyanins in the finger millet seed testa.

## Discussion

Finger millet is the third Chloridoid species to have a reference-quality chromosome-resolved genome assembly generated. At 1.1 Gb (C), the genome assembly is 27% smaller than the most recent estimated sizes of the finger millet genome (~1.5 Gb)[3,4], and in line with earlier published genome assemblies (1.2 Gb)[3,4]. Comparative information, as well as a BUSCO value of 96.9% indicate that the assembly is largely complete. Although the tetraploid finger millet genome is considerably larger than that of the other Chloridoids with chromosome-level

genome assemblies, the diploid *Oropetium thomaeum* (C = 244 Mb)[15] and the tetraploid *Eragrostis tef* (C = 576 Mb)[16], it is the highest quality Chloridoid assembly generated to date and, with a contig N50 of 15.3 Mbp, a 50- to 600-fold improvement in contiguity over earlier published finger millet assemblies (Supplementary Table 1). Similar to teff, finger millet is a relatively young allotetraploid that arose around 1.3 MYA (1.1 MYA for teff)[16], accounting for the high level of retained homoeologs in this crop. Genome instability immediately following allopolyploidization is common (reviewed by Mason and Wendel[29]), but homoeologous exchanges can also occur in well-established polyploids with strong pairing control such as wheat[48]. From the alignment of *E. indica* reads to the finger millet genome assembly, we identified two reciprocal translocations between homoeologous chromosomes, a 6A/6B translocation that predates domestication and a 9A/9B translocation that predates the spread of finger millet from its center of domestication. The 9A/9B translocation breakpoint lies within an intron and resulted in a hybrid gene, a pattern that has been observed in several polyploids[30]. Homoeologous exchanges, if leading to new gene functions, may provide a selective advantage leading to their fixation in the population.

The domestication of finger millet, likely in the Ethiopian highlands, and its subsequent dispersal, first to the African lowlands and then to India, gave rise to geographically stratified cultivated germplasm pools. Within the lowland African pool, three subpools can be recognized. The main driver of the population structure is not genome-wide diversification, but rather the shared presence of wild alleles in pericentromeric regions, which can be caused by a number of factors. Mutation rates may be lower in the pericentromeric regions in finger millet, as has been observed in maize[49]. Genes in those regions may also be under strong purifying selection, leading to a slower evolution of the pericentromeric regions[50,51]. Further, due to the low recombination rates in these regions, these segments will likely be inherited in their entirety. Post-domestication introgression of wild segments in pericentromeric regions cannot be excluded, at least in regions where subsp. *coracana* and *africana* grow in sympatry, but would be expected to occur at low frequency due to the low recombination rates near centromeres[52]. Because subsp. *africana* is not endemic in Asia, the presence of wild segments across the Asian subpopulation suggests that the introduced germplasm pool was likely small and limited in its geographic origin. None of the African germplasm pools carry the pericentromeric 5A wild region that characterizes Asian finger millet lines, precluding the use of this characteristic to determine the origin of the Asian germplasm pool. Fst and Dxy values, however, point at pop1-pop2 as the African source population, which agrees with a lowland African origin. As crosses between Indian and African material become more common, continued retention of the 5A segment in Asia or, conversely, loss in Africa, could be an indicator that the presence of this region, and by extension the other subpopulation-specific regions, have selective advantages in defined geographic locations or under specific finger millet management practices.

While the generation of the genome assembly and the germplasm diversity analysis are part of an effort to develop genomic resources for finger millet to facilitate trait mapping and identification of causal genes in aid of breeders, these resources will also benefit the scientific community at large. Understudied species are almost certain to carry unique variants that will enhance our knowledge on genes and pathways. As a case study, we exploited the generated resources to identify and characterize the gene responsible for the variation in anthocyanin levels observed in stigma and anthers in finger millet germplasm, a visual marker widely used by breeders to identify F$_1$ hybrids. Loss of flavonoid production can result from mutations in regulatory genes, typically a complex of bHLH, WD40 and Myb transcription factors[53], or in structural genes. The key gene that led to the loss of anthocyanin production in reproductive organs in the majority of analyzed finger

millet accessions, and likely in internodes where the trait is less penetrant, is *PP*. Our data strongly suggest that *PP* is a MYC-bHLH transcription factor orthologous to the homoeologous maize genes *R1* and *B1*, and the *Arabidopsis* duplicated genes *GL3* and *EGL3*. GL3 and EGL3 have partially redundant regulatory functions in, amongst other traits, anthocyanin production, while bHLH family member TT8 regulates seed coat tannin and mucilage production[45]. The maize bHLH transcription factor R1, but not its homoeolog B1, can compensate for all three *Arabidopsis* mutations[54], indicating subfunctionalization of R1 and B1. Maize seed does not have condensed tannins, and the maize TT8 ortholog, IN1, negatively affects anthocyanin production by competing with R1[47], another example of sub- or, potentially, neofunctionalization of a gene following species diversification. In finger millet, only a single bHLH transcription factor, *PP*, is associated with anthocyanin production in stigma and anthers. The homoeolog of *PP* has accumulated high-impact mutations in the coding region, which either caused functional inactivation or arose subsequent to an earlier inactivation event, potentially during allopolyploidization, that freed the gene from selective constraints. Allopolyploidization is known to accelerate genome evolution, including through gene silencing resulting from transposable element mobilization and epigenetic modifications[55]. The *PP* gene underwent at least two independent partial deletions, one in the wild accession MD-20 and one in cultivated germplasm, that are associated with the loss of anthocyanin production. Loss of purple coloration of reproductive organs is expected to be selectively neutral or even advantageous in an inbreeding species that is not dependent on pollinator attraction[56]. Finger millet accessions lacking anthocyanins can, however, still produce proanthocyanidins, which have been ascribed protective functions[57], in the seed coat. This suggests that, as in *Arabidopsis*, the two pathways are controlled by different bHLH transcription factors. The finger millet orthologs of *Arabidopsis TT8* and maize *IN1* are the likely candidate regulators for seed proanthocyanins. White-seeded accessions and also a few brown-seeded accessions have been reported to lack proanthocyanidins[38], and further investigation is needed to establish the relationship between seed coat color and seed flavonoid types and levels. Knocking out the chalcone synthase genes on homoeologous chromosomes 9A and 9B likely abolishes both anthocyanin and proanthocyanidin production in the reproductive organs and seed. However, these function-inactivating mutations were not found in two white accessions that had purple stigma and anthers. Our finding of separate regulation of anthocyanin and proanthocyanidin biosynthesis contradicts the popular belief, based on a 1931 assertion[58], that white grains can be produced only on finger millet plants that lack anthocyanins. Final proof that *PP* controls purple coloration of stigma and anthers in finger millet while the paralogs ELECO.r07.7AG0560560 and ELECO.r07.7BG0592150 control seed proanthocyanidin production will have to await the development of a routine transformation system in finger millet.

## Methods

### Genome sequencing

Seed from a single self-pollinated plant of *E. coracana* subsp. *coracana* accession KNE 796-S was planted in trays and grown under artificial lighting (14 h days) at 27 °C. Seedlings were harvested approximately one month postemergence. For DNA extractions[59], frozen leaves were ground to a fine powder in liquid nitrogen in a precooled mortar and gently extracted in a CTAB buffer that included proteinase K, PVP-40 and β-mercaptoethanol for 1 h at 50 °C. Proteins were removed by two gentle extractions with 24:1 chloroform:isoamyl alcohol. After the addition of 1/10 volume of 3 M KAc, DNA was precipitated with isopropanol. The DNA pellet was washed with 70% ethanol, air dried for 20 min and dissolved in 1x Tris-EDTA (TE)-buffer at room temperature. The DNA was sent on dry ice to HudsonAlpha for library construction and sequencing.

Sequencing was done at the HudsonAlpha Institute in Huntsville, Alabama, using a whole genome shotgun sequencing strategy and standard sequencing protocols. Illumina reads were sequenced using the Illumina X10 and NovoSeq6000 platforms, and the PACBIO reads were sequenced using the SEQUEL I platform. One 400 bp insert 2×150 Illumina fragment library (94.59x) was sequenced along with one 2×150 Dovetail Hi-C library (164.2x). Prior to assembly, Illumina fragment reads were screened for PhiX contamination. Reads composed of >95% simple sequence and reads <50 bp after trimming for adapter and quality ($q < 20$) were removed. For the PACBIO sequencing, a total of 28 PB chemistry 2.1 chips (10 h movie time) were used.

## Genome assembly and construction of pseudomolecule chromosomes

We generated a reference-quality chromosome-resolved assembly with a contig N/L50 of 25/15.3 Mb and a scaffold N/L50 of 9/61.3 Mb for the cultivated Kenyan finger millet accession KNE 796. To build the reference genome, 5,398,480 PACBIO long-read sequences (152.47 Gb of raw sequence reads; 84.71x genome coverage; 6981 bp average read length) were assembled using MECAT v1.4[60] into 1058 scaffolds (1058 contigs). This first assembly had a contig N50 of 12.1 Mb, 228 scaffolds larger than 100 Kb and covered 1129.7 Mb. Polishing was accomplished with QUIVER v2.1[61]; 280 homozygous single nucleotide polymorphisms (SNPs) and 18,709 homozygous insertion-deletions (INDELS) were corrected based on 695,953,098 (94.59x) Illumina short-read sequences (average insert size: 400 bp). Final scaffolding was accomplished using 1,214,604,949 (164.2x) Hi-C reads integrated with the JUICER v1.8.9[62] pipeline, which clusters Hi-C contacts into groups, and a 4400-marker genetic map[28]. Four misjoins, characterized as discontinuities in a linkage map, were identified and resolved, and 178 joins were made to assemble the contigs into 18 chromosomes. Duplicated overlapping sequences in 36 adjacent contig pairs were collapsed. The resulting broken contigs were then oriented, ordered and joined together into 18 chromosomes using both the map and the Hi-C data. Each chromosome join was padded with 10,000 Ns. Significant telomeric sequence was identified using the (TTTAGGG)$_n$ repeat, and care was taken to ensure that it was oriented correctly in the production assembly. The remaining scaffolds were screened against bacterial proteins, organelle sequences and GenBank nr, and removed if found to be a contaminant. After forming the chromosomes, it was observed that some small (<20 Kb) redundant sequences, which are exact duplicated artefacts of assembly, were present on adjacent contig ends within chromosomes. To fix this problem, adjacent contig ends were aligned to one another using BLAT[63], and duplicate sequences were collapsed to close the gap between them. Finally, homozygous SNPs and INDELs were corrected in the release consensus sequence using ~65x of Illumina reads (2×150, 400 bp insert) by aligning the reads using BWA-mem v0.7.17-r1188[64] and identifying homozygous SNPs and INDELs with the Genome Analysis Toolkit's (GATK v3.6-0g89b7209) UnifiedGenotyper tool[65]. It is important to note that flow cytometry estimates have projected the cultivated finger millet genome to be in the size range 1.5 – 1.9 Gb[4,66]. However, neither of the existing genomes[3,4] were able to scaffold more than 1.3 Gb of sequence. Our genome at 1.1 Gb is smaller than previous genomes. This is somewhat surprising given the technology employed here is far better at representing low-complexity pericentromeric sequences. The reduced size of our genome may result from differences in sequencing technologies, including collapsing or expanding of heterozygous regions. It may also be that KNE 796-S has a biologically smaller genome as has been observed for wild subsp. *africana* accessions, which have flow cytometric DNA estimates in the range 1.2 – 1.6 Gb[3,66].

## Repeat annotation

Repeat annotation combined de novo and homology-based annotation of repeats[16]. Simple sequence repeats were identified and masked in the genome assembly using GMATA v2.3[67]. Initially, structure-based identification of full-length transposable elements (TEs) was conducted. Long terminal repeat retrotransposons (LTR-RTs) were mined from the genome assembly with LTR-finder v1.1[68] and LTRharvest v1.6.2[69], and high-confidence elements were distilled from this set using LTR_retriever v2.9.0[70]. Short interspersed nuclear elements (SINEs) were identified with SINE-scan v1.1[71], long interspersed nuclear elements (LINEs) with MGEscan-nonLTR v2.0[72], miniature inverted-repeat transposable elements (MITEs) and other DNA elements with terminal inverted repeats (TIRs) using MITE-Hunter v1.0[73] and MITE Tracker v1.0[74], and Helitrons with HelitronScanner v1.0[75]. All TEs were classified according to the nomenclature system for transposons[76] and their annotation validated against Repbase[77]. The structurally identified TEs were merged with Repbase and used as a custom library to identify full-length and truncated TE elements through a homology-based search with RepeatMasker v4.0.7 (http://www.repeatmasker.org) using the unmasked assembly as input. The distribution of LTR-RT families with at least five intact copies was calculated from the gtf file generated by RepeatMasker v4.0.7.

## Gene annotation

The annotation pipeline combined three methods for structural gene annotation in plants: protein homology, expression data based and ab initio prediction[78].

Homology-based annotation used available *Triticeae* protein sequences (UniProt (05/10/2016)), which were mapped to the nucleotide sequences of the finger millet pseudomolecules using GenomeThreader v1.7.1[79]. RNASeq datasets were mapped to the genome assembly using HISAT2 (v2.0.4, parameter −dta)[80] and subsequently assembled into transcript sequences by StringTie (v1.2.3, parameters -m 150 -t -f 0.3)[81]. RNASeq datasets of PRJNA377606 and PRJNA648385 were downloaded from NCBI. Transcripts were combined using Cuffcompare v2.2.1[82] and subsequently merged with StringTie (version 1.2.3, parameters−merge -m 150). Transdecoder v3.0.0 (https://github.com/Transdecoder/Transdecoder) was used to find potential open reading frames and to predict protein sequences.

Ab initio annotation using Augustus v3.3.2[83] was carried out to further improve the structural gene annotation. In order to minimize over-prediction, hint files using the above-mentioned RNASeq, protein evidence and TE predictions were generated. The wheat model was used for prediction.

All structural gene annotations were joined by EVidenceModeller v1.1.1[84], with weights adjusted according to the input source. Finally, candidates were classified into high or low-confidence categories. High-confidence models are models that are complete (with START/STOP) and with a query/subject coverage greater than 75% when compared to reference proteins.

Functional annotation of predicted protein sequences was done using the AHRD pipeline (https://github.com/groupschoof/AHRD). Completeness of the predicted gene space was measured with Benchmarking Universal Single-Copy Orthologs (BUSCO; v3.02, orthodb9) (https://gitlab.com/ezlab/busco).

## Dating the divergence of the A and B subgenomes

Homoeologous genes with a 1-to-1 relationship between the A and B subgenomes ($n = 16,448$) (Supplementary Data 9) were used to estimate the number of synonymous substitutions per synonymous site ($K$s). Homoeologous sequence pairs were aligned using ClustalO v1.2.2[85]. Multiple sequence alignments of proteins were converted to codon alignments of the corresponding DNA sequences using PAL2-NAL v14[86]. PAML v4.10.3[87] was used to estimate $K$s using the Nei and Gojobori method[87]. Based on the median $K$s, the divergence time ($T$)

was estimated using the mutation rate (*r*) of grasses ($6.5 \times 10^{-9}$ substitutions per synonymous site per year)[27]. *Ks* values > 1 were removed to eliminate saturated synonymous sites.

$$T = Ks/(2r) \qquad (1)$$

## Dating the finger millet tetraploidization event

The insertion dates of LTR elements were estimated by the degree of divergence of their two LTRs using the formula

$$t = d/2r \qquad (2)$$

where *t* is the insertion date, *d* is the evolutionary distance between two LTRs of an element and *r* is the rate of base substitution. The value of r used in this study was $1.3 \times 10^{-8}$ substitutions per site per year, as proposed by Ma and Bennetzen[88]. We used the average transposition date of the 25% youngest elements in three B-genome-specific families with more than 80 intact elements as a measure for the date of the tetraploidization event.

## Assessment of subgenome dominance

To assess whether one of the subgenomes is dominant in terms of gene expression, we compared transcript levels of homoeologous gene pairs with a 1:1 relationship in the two subgenomes (Supplementary Data 9). RNASeq reads for KNE 796 were downloaded from NCBI SRA (BioProject PRJNA377606; Runs SRR5341138-SRR5341148. The raw reads were trimmed to remove low-quality sequences (Phred score <33) using the paired-end mode of Trim Galore v0.45 (https://github.com/FelixKrueger/TrimGalore). The trimmed reads were aligned against the finger millet KNE 796-S reference genome with HISAT2 v2.1.0[80]. The aligned reads were assembled into transcripts with guidance of the KNE 796-S annotation using StringTie v2.1.1[81]. Transcripts were merged across samples and used as a reference for transcript quantification. Quantified GTF files were converted to a gene count matrix using the following formula[89]

$$Reads\_per\_transcript = coverage * transcript\_len/read\_len \qquad (3)$$

Expression difference tests were conducted within pairs of homoeologous genes with a 1-to-1 relationship (Supplementary Data 12) using a PROC GLM MANOVA for all tissues in SAS v9.4 (SAS Institute, 2012, Cary, NC, USA). Subgenome was used as a predictor variable. ANOVAs were subsequently conducted for each tissue type independently. For comparisons, we used a Tukey's LSmeans multiple comparison adjustment.

## Comparative analyses

Synteny maps between the finger millet A and B genome chromosomes were generated with GENESPACE v0.9.3[90], which uses orthologs generated from the peptide annotations and position coordinates for the gff files.

For comparative analyses across species, the proteins corresponding to the primary transcripts in *Oryza sativa* v7.0[18], *Oropetium thomaeum* v1.0[15], *Eragrostis tef* v3[16] and *Sorghum bicolor* v3.1.1[17] were downloaded from the sources listed in Supplementary Table 13. The proteins annotated in the finger millet A and B genomes were used as queries in separate BLASTP searches against the proteins downloaded from the four grass species, and the top two hits with an E-value < $1e10^{-5}$ were recorded. The top finger millet protein hits and second-best hits for each species were used for syntenic block detection using the software MCScanX[91] with a match score of 50, match size of 5, gap penalty of −1, overlap window of 5, E-value of $1e^{-5}$, and max gaps of 25. The output of MCScanX as well previously established

comparative relationships[19,20] were used to manually generate a circle diagram depicting grass relationships.

## Resequencing of finger millet accessions

A total of 20 diverse accessions identified morphologically as *E. coracana* subsp. *coracana*, four identified as *E. coracana* subsp. *africana* and two as *E. indica* (AA genome) were resequenced on an Illumina platform. An additional 15 presumed *E. coracana* subsp. *coracana* and seven *E. coracana* subsp. *africana* accessions were sequenced on an Ion Proton Platform. The list of accessions, together with their source and country of origin, is provided in Supplementary Data 3.

For Illumina sequencing, DNA extractions were conducted from leaf tissue using a CTAB method[59]. A total of 1.5 µg of DNA was sheared to 350 basepairs using a Covaris-focused ultrasonicator LE-Series LE220. Unamplified libraries were constructed in 96-well format using an Illumina TruSeq DNA PCR-free high throughput kit and standard protocols. Sequencing was performed on an Illumina NovaSeq 6000 instrument using a NovaSeq6000 S4 Reagent Kit and 300 cycles.

For samples sequenced on the Ion Proton platform, genomic DNA was extracted from 2-week-old seedlings using a modified CTAB protocol. Libraries were prepared using the Ion XpressTM Plus Fragment Library Kit (Life Technologies, CA, USA) according to the manufacturer's recommendations. Briefly, 1 µg of RNA-free high molecular weight genomic DNA was enzymatically sheared and end-repaired. Following adapter ligation and nick repair, the libraries were size-selected (300 – 450 bp) using a Pippin Prep (Sage Science). The size-selected DNA was amplified for eight cycles as recommended for 600 base-read libraries, purified using 375 µL Agencourt Ampure beads, eluted in 50 µL low-EDTA TE buffer and quantified using an Agilent High Sensitivity DNA Kit (5067-4626) on an Agilent 2100 Bioanalyzer (Agilent Genomics). Qualified DNA libraries were then loaded onto an Ion Chef™ Instrument (Life Technologies, USA) for template enrichment with an Ion PI™ Hi-Q™ Chef kit (A27198) and Ion PI™ chip v2 loading. Once the chips were loaded by the Ion Chef™ Instrument, the DNA libraries were sequenced on an Ion Proton™ sequencer with PI™ Hi-Q™ Sequencing 200 Kit (A26433) chemistry (Life Technologies, USA) using 520 flows.

## Genotyping-by-sequencing of finger millet accessions

A total of 294 *Eleusine coracana* and three *E. indica* accessions from 11 African and four Asian countries, including nine accessions of unknown origin, were subjected to GBS (Supplementary Data 3). From these accessions, 282 had been classified as subsp. *coracana* (cultivated) and 12 as subsp. *africana* (wild).

Genomic DNA was isolated from leaf tissue using a CTAB method[59] and used for GBS library preparation[28]. GBS libraries with a concentration higher than 5.0 ng/µL were pooled in sets of 200 for sequencing on an Illumina NextSeq platform (2×150 bp) at the Georgia Genomics and Bioinformatics Core (GGBC). The GBS reads in this study were generated on five flow cells with each flow cell containing additional samples not related to the research described here. For quality control, 12 libraries were run on two different flow cells. We also made duplicate libraries for 18 samples and sequenced them on separate flow cells. After it was ascertained that duplicate samples grouped together phylogenetically, scores across duplicates were merged[28].

## Single nucleotide polymorphism calling

Processing of GBS and resequencing reads, and SNP calling followed the reported workflow[28] and is described in brief below. After the removal of adaptor sequences and trimming of reads for quality and length, GBS and resequencing reads were aligned separately against the KNE 796-S finger millet genome assembly using Bowtie2 v2.4.1[92]. SNP calling was done separately for the GBS and resequencing reads using the Genome Analysis Toolkit (GATK) v3.4[65]. GATK results within each dataset were preliminary filtered to remove all SNPs with an allele

frequency <2%, a quality-by-depth (QD) < 10 and >30% of missing data. The remaining SNPs in the resequencing dataset were functionally annotated using snpEff v4.3[93].

In one of the GBS sequencing runs, a consistent, non-random error at the beginning of the reverse reads was observed, and all SNPs present in the first 25 bp of these reads were also removed. The shared SNPs between the GBS data and the resequencing data were combined and further filtered to remove SNPs with >20% missing data and a minor allele frequency <5%. Because finger millet is an inbreeding allopolyploid species, and heterozygous SNPs were likely caused by co-mapping of reads from homoeologous loci, we also removed SNPs that were heterozygous in ≥10% of the samples. Accessions with more than 20% of missing SNPs were also removed. All SNPs with a read depth ≥8X on a per-sample basis were scored as A (reference allele), B (alternate allele) or H (heterozygous).

### Genetic diversity analysis

The shared GBS/resequencing SNPs generated across all analyzed accessions were split into two datasets, with one set containing A-genome SNPs and all accessions, and the other set containing B-genome SNPs and all accessions except *E. indica* (a diploid A-genome species). In addition, because of the presence of homoeologous 6A/6B and 9A/9B translocations in KNE 796-S, we replaced the SNPs in the 6A and 9A regions with the SNPs in the 6B and 9B regions of KNE 796-S in the A dataset, and vice versa. Population structure in the two datasets across the genome as well as on a chromosome-by-chromosome basis was analyzed using a Bayesian clustering algorithm implemented in STRUCTURE v2.3.4[94], allowing admixture and using a burn-in period of 100,000 replications and 1,000,000 Markov chain Monte Carlo (MCMC) iterations. The optimal number of clusters or subpopulations (K) was inferred using the ΔK method[95]. Individuals with ≤75% membership to any single subpopulation were considered to be admixed. Principal Coordinates Analysis (PCoA) and pairwise estimates of genetic differentiation (Fst) between subpopulations as defined by STRUCTURE were performed using GenAlEx v6.501[96]. Dxy values (Nei[97], equation 10.20) were calculated under DnaSP6 v6.12.03[98]. The PCoA analysis was based on 3000 A-genome SNPs and 3000 B-genome SNPs, selected from the total SNP set to maximize diversity in each genome using Core Hunter v2.0[99], with default weights of Mean Rogers' distance and Shannon diversity of 70% and 30%, respectively.

### QTL mapping of purple anther and stigma color

The MD-20 (*E. coracana* subsp. *africana*) x Okhale-1 (*E. coracana* subsp. *coracana*) population had previously been genetically mapped using GBS[28]. However, because SNP calling by Qi and colleagues[28] had been done in the absence of a reference genome, we recalled SNPs from the previously generated GBS reads (https://www.ncbi.nlm.nih.gov/sra/?term=SRP136342) using the KNE 796-S v.1.0 assembly described here as reference. The updated genetic maps were assembled using a combination of MSTMap, MapMaker and in-house scripts[28]. We manually reordered markers that were not separated by a solid recombination event (double recombination events were ignored) according to their order in the genome assembly. We also removed samples 8, 25, 101 and 135 because of the large number of predicted double recombination events in these lines, a common outcome of seed or DNA contamination.

Anther and stigma color, which cosegregated in the $F_2$ population, were scored as a binary trait (white stigma/yellow anthers = 1, purple stigma/purple anthers = 2) in 122 of the 129 genotyped $F_2$ progeny. QTL were identified using composite interval mapping with a walk speed of 1 cM in R/QTL v1.52[100]. Significance thresholds (α = 0.05) for the QTL for anther/stigma color were determined by conducting 1000 permutations. The percentage variation explained (PVE) was calculated using the 'makeqtl' and 'fitqtl' functions in R/QTL[100].

Microscopy of finger millet ovules of MD-20 and Okhale-1 was conducted with a Leica DVM6 light microscope. Panicles were harvested the day of microscopy at the S3 grain-filling stage of finger millet spike development[101], and ovules were imaged immediately upon dissection. Images were processed using the Leica Application Suite X software (3.0.12.21488).

### Measurement of condensed tannins

Measurement of condensed tannins followed the vanillin/HCl method[102]. Approximately two grams of finger millet grains were manually ground into fine flour. A total of 20 mg of flour was suspended in 1 mL methanol, shaken at room temperature for 1 h, and then centrifuged for 10 min. at 1000 g. Two-hundred microliter of supernatant was added to 1.8 mL vanillin reagent (4% (w/v) vanillin and 10% (v/v) HCl in methanol) and the mix was incubated at 30 °C for 20 min. Absorbance at 500 nm was measured on a Biotek Cytation 5 Cell Imaging Multimode Reader. To obtain the absolute amount of condensed tannins, a standard curve was generated with 0, 30, 60, 90 and 120 mg/L of catechin dissolved in methanol. Sample blanks consisted of 1.8 mL vanillin reagent with 200 μL methanol.

### Candidate gene identification

Genes located in the chromosome 4A QTL interval delineated by markers that were significantly ($p < 0.05$) associated with anther/stigma color were downloaded from the annotated (v1.1) finger millet KNE 796-S v1.0 genome assembly (https://phytozome-next.jgi.doe.gov/info/Ecoracana_v1_1). Genes located in the QTL interval with a description or GO annotation in Phytozome indicating involvement in the anthocyanin pathway were analyzed for variation present between Okhale-1 and MD-20 based on the whole-genome resequencing data. Because finger millet is an allotetraploid and purple anther/stigma color is dominant, we reasoned that the 4B homoeolog of the causal gene to the purple pigmentation should also be non-functional in the non-pigmented parent (MD-20). Furthermore, because no QTL was identified on chromosome 4B, the 4B homoeolog in the pigmented parent (Okhale-1) should also be non-functional. Genes in the QTL interval with this pattern of high-impact mutations were considered very strong candidates for anther/stigma color.

### PCR validation and assessment of variants and the 9 A/9B translocation

The primers used in the PCR reactions are listed in Supplementary Table 14. Reactions consisted of 30 to 50 ng of template DNA, 200 μM dNTPs, 0.4 μM forward and reverse primers, 1.5 mM MgCl$_2$ (2.5 mM MgCl$_2$ for primer set 4B338780F/4B338780R) and 0.8 U Taq Flexi Polymerase in 25 μL 1X Go Taq Flexi Buffer (Promega Corporation). PCR conditions were denaturation at 95 °C for 5 min followed by 35 cycles (40 cycles for primer sets 9AT_F/9A_R2F and 9BT_F/9A_R2F) of denaturation at 95 °C for 30 sec, annealing at the appropriate temperature (Supplementary Table 14) for 30 sec, and extension at 72 °C for 30 to 90 sec depending on the primer set (Supplementary Table 14), and a final extension at 72 °C for 5 min.

### Phylogenetic analysis of select MYC-bHLH transcription factors

The *Arabidopsis* MYC-bHLH transcription factors GLABRA3 (GL3), ENHANCER OF GLABRA3 (EGL3) and TRANSPARENT TESTA 8 (TT8) were used as queries in BLASTP searches against the annotated proteomes of finger millet (*Eleusine coracana* v1.1), maize (*Zea mays* RefGene_V4), foxtail millet (*Setaria italica* v2.2) and rice (*Oryza sativa* Kitaake v3.1) present in Phytozome (https://phytozome-next.jgi.doe.gov/). Top hits in the four species that were identified in all three searches were aligned with MUSCLE using default parameters in Jalview v2.11.1.5[103]. The homoeologous proteins ELECO.r07.2AG0117910 and ELECO.r07.2BG0171500, which showed the highest homology to TT8 outside the finger millet top hits, were used as outgroup. A maximum likelihood tree was built from the protein alignment using MEGA v11.0.13[104] (parameters: Model = Jones-Taylor-Thornton (JTT) model,

Rates among sites = Uniform Rates, Gaps/Missing data treatment = Complete deletion, ML Heuristic Methods = Nearest-Neighbor-Interchange, Branch Swap Filter = none, 1000 bootstraps).

## Reporting summary

Further information on research design is available in the Nature Portfolio Reporting Summary linked to this article.

## Data availability

The datasets generated during this study are available from NCBI's Sequence Read Archive (PACBIO, Illumina sequencing reads and annotated genome assembly for KNE 796-S: BioProject PRJNA838475; Illumina sequencing reads for other finger millet accessions: BioProject PRJNA838475; Ion Proton sequencing reads: BioProject PRJNA876392; GBS reads: BioProject PRJNA870151). The annotated KNE 796-S genome assembly is also available from Phytozome [https://phytozome-next.jgi.doe.gov/info/Ecoracana_v1_1]. The following datasets retrieved from NCBI's SRA were used as part of the study: RNAseq data: PRJNA377606 and PRJNA648385; GBS data of MD-20 x Okhale-1 mapping population: SRP136342. Seed of the sequenced finger millet accession, KNE 796-S, has been deposited in the U.S. National Plant Germplasm System (NPGS) under accession number PI 702583 [https://npgsweb.ars-grin.gov/gringlobal/accessiondetail?id=2141930]. Distribution of the other finger millet germplasm may be restricted due to country-of-origin specific regulations and seed stock limitations. Please contact K.M. Devos (kdevos@uga.edu) for further information. Seed from non-restricted germplasm will be distributed, pending availability and unforeseen circumstances, within 3 weeks of receiving the request. Source data are provided with this paper or are available from Figshare [https://doi.org/10.6084/m9.figshare.22762430][105].

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

## Acknowledgements

This research was supported by awards from the National Science Foundation – Basic Research to Enable Agricultural Development (NSF-BREAD) (award #1543901 to K.M.D., M.C.S., D.O., M.M.D., K.T. and S.d.V.), BioInnovate Africa (award to K.T., E.E.M., S.d.V., K.M.D. and M.M.D.) and Consortium of International Agricultural Research Centers (CGIAR) – Dryland Cereals partnership funds. D.S. was supported by a United States-India Educational Foundation (USIEF) Fulbright Nehru Senior Research Fellowship (2013-14). The work conducted by the US Department of Energy Joint Genome Institute is supported by the Office of Science of the US Department of Energy under Contract No. DE-AC02-05CH11231. We thank J.L. Bennetzen for critically reading the manuscript.

## Author contributions

K.M.D. designed and coordinated the research, and participated in data analysis and interpretation; S.d.V. coordinated discussions on varietal choice for reference genome generation; H.F.O provided the seed of KNE 796-S; J.S. coordinated and J.G. conducted the PacBio and Illumina genome sequencing; J.J. and C.P. generated the genome assembly; M.C.S. and K.J. contributed to the assembly quality analysis; T.L. and K.F.X.M. performed the gene annotation and BUSCO assessment of the annotation; H.W. and X.W. conducted the repeat annotation and repeat-based analyses; A.S. conducted the divergence analysis; D.A.O., S.d.V. and S.D. co-coordinated germplasm provision; K.T., E.E.M., M.M.D., D.L. and S.J.M. provided finger millet germplasm, DNA and/or morphological characteristics; L.M-B. and D.S. conducted GBS analyses; B.A.B., M.C.S and K.J. conducted the population structure analyses; P.Q. and J.L. analyzed allele ratios; D.A.O. coordinated RNAseq data generation and Ion Proton resequencing; B.S. and D.M.G. generated RNAseq data; T.H.P. conducted statistical analyses; M.M.D. generated the MD-20 x Okhale mapping population and conducted the phenotyping for anther and stigma color; G.G.S conducted the translocation analyses and PCR validations; P.Q. conducted the comparative analyses; J.Z. and P.Q conducted the gene candidate analyses; H.Wr. conducted the microscopy; K.M.D. drafted the manuscript with input from co-authors; All co-authors approved the manuscript.

## Competing interests

The authors declare no competing interests.

## Additional information

[1]Institute of Plant Breeding, Genetics and Genomics, University of Georgia, Athens, GA 30602, USA. [2]Department of Crop and Soil Sciences, University of Georgia, Athens, GA 30602, USA. [3]Department of Plant Biology, University of Georgia, Athens, GA 30602, USA. [4]Department of Plant Pathology, University of Georgia, Griffin, GA 30223, USA. [5]International Crops Research Institute for the Semi-Arid Tropics (ICRISAT) – Eastern and Southern Africa, P.O. Box 39063-00623 Nairobi, Kenya. [6]Departments of Computer Science, Biology and Genetic Medicine, Johns Hopkins University, Baltimore, MD 21218, USA. [7]Oromia Agricultural Research Institute, P.O. Box 81265 Addis Ababa, Ethiopia. [8]Plant Genome and Systems Biology, German Research Center for Environmental Health, Helmholtz Zentrum München, 85764 Neuherberg, Germany. [9]Genome Sequencing Center, HudsonAlpha Institute for Biotechnology, Huntsville, AL 35806, USA. [10]Department of Genetics, University of Georgia, Athens, GA 30602, USA. [11]ICRISAT, Patancheru 502 324 T.S., India. [12]Department of Biochemistry and Biotechnology, Pwani University, Kilifi 80108, Kenya. [13]Pwani University Biosciences Research Center (PUBReC), Kilifi 80108, Kenya. [14]Department of Crop and Soil Science, Maseno University, P.O. 333 Maseno, Kenya. [15]US Department of Energy Joint Genome Institute, Lawrence Berkeley National Laboratory, Berkeley, CA 94720, USA. [16]School of Life Sciences Weihenstephan, Technical University of Munich, 85354 Freising, Germany. [17]Mikocheni Agricultural Research Institute, P.O. Box 6226 Dar Es Salaam, Tanzania. [18]ICRISAT, Matopos Research Station, P.O. Box 776 Bulawayo, Zimbabwe. [19]BGI-Shenzhen, Beishan Industrial Zone, Yantian District, Shenzhen 518083, China. [20]Institute of Biotechnology, Addis Ababa University, Addis Ababa, Ethiopia. [21]Bio and Emerging Technology Institute, Addis Ababa, Ethiopia. [22]Present address: Department of Horticulture, University of Georgia, Athens, GA 30602, USA. [23]Present address: Ethiopian Agricultural Transformation Agency, Addis Ababa, Bole, Ethiopia. [24]Present address: UR Ventures, University of Rochester, Rochester, NY 14627, USA. [25]Present address: ICAR-Central Research Institute for Jute and Allied Fibers, Kolkata, West Bengal 700120, India. [26]Present address: Hytech Seed India Pvt. Ltd., Ravalkol Village, Medcahl-Malkajgiri Dist–, 501 401 Hubballi, T.S, India. [27]Present address: Biotechnology Society of Tanzania, P.O. Box 10257 Dar es Salaam, Tanzania. [28]Present address: Agricultural Genomics Institute at Shenzhen, Chinese Academy of Agricultural Sciences, Shenzhen 518120, China. [29]These authors contributed equally: Katrien M. Devos, Peng Qi. ✉e-mail: kdevos@uga.edu

