## [Peer Review File · Nature Communications]

Genome analyses reveal population structure and a purple stigma color gene candidate in finger milletEditorial Note: This manuscript has been previously reviewed at another journal that is not operating a transparent peer review scheme. This document only contains reviewer comments and rebuttal letters for versions considered at Nature Communications.

Reviewers' Comments:

Reviewer #4:

Remarks to the Author:

Devos et al. report a chromosome scale genome assembly for the allotetraploid cereal finger millet and conducted downstream population and quantitative genetics related to domestication traits. I did not review the original version of this manuscript, but I have read this updated draft and the authors comments to the previous reviewers. I read this paper with interest, and think it represents a useful resource to help accelerate breeding of this important but underutilized cereal. Although the manuscript is generally improved, I think there are still a few issues that need to be addressed before publication.

Mainly, I do not think the evidence for identifying the gene underlying purple stigma to be very strong, and on a deeper level, I do not understand the significance of this trait to finger millet breeding or domestication. In line 313 the authors claim this phenotype is used for breeding, but they do not provide a source for this. Stigma color is also independent to purple/brown seeds, so I am not sure what the agronomic relevance is for this. Perhaps this is a misunderstanding on my part, but readers will also probably not be experts in finger millet breeding, so a clearer connection to this trait and improvement is needed. Importantly, the authors also state that only 15 out of 22 accessions with the functional PP allele displayed purple coloration and hypothesize that the other seven accessions have other non-functional alleles or deleterious mutations in other genes of the anthocyanin biosynthesis pathway. They provide some evidence for this in their revision by sequencing one of the white lines and identifying two possible loss of function mutations in a chalcone synthase gene. This is an interesting hypothesis and set of analyses, but I disagree that this overwhelmingly supports the MYC-bHLH transcription factor to be the causal gene underlying the QTL that controls purple stigma, as stated by the authors. Including these analyses in the paper is certainly fine, but in my opinion, these results do not 'reveal the genetic basis of purple stigma color' as claimed in the title and abstract, and the paper should be reframed as such to reflect this uncertainty. The authors specifically state this trait evolved at least twice through loss of functions in MYC-bHLH, but no downstream functional validation was done to verify this. I understand the difficulties with transformation of this species, but the use of phrases such as 'likely', or 'candidate' would be sufficient to address this major concern.

Minor comments:

Line 429. This is not true, and a number of other chloridoid species have been sequenced including several *Zoysia* sp. *Eragrostis curvula*, *Eragrostis nindensis*, and several *Sporobolus* species, among probably others. Although lower quality, other finger millet genomes are also available but are still not really mentioned here.

Line 435. I'm not sure this is accurate. Contig N50 is becoming a meaningless statistic by itself without factoring in other characteristics such as genome size or chromosome number. For instance, a larger genome with a lower chromosome number could have a higher N50 but be more fragmented than a smaller genome.

Reviewer #5:

Remarks to the Author:

Review of the article Devos et al. genome analyses reveal the genetic basis of population structure and purple stigma color in finger millet. The article is highly relevant to agriculture application, adaptation of agriculture to climate and study of neglected crops.

The article present a new assembly of the finger millet genome with better continuity, an analysis of A and B chromosome and a population structure shape by pericentromerics region, finally some likely candidate for purple color variation in the flower.

Minor comments

Overall, the authors answer to the comment in a satisfying way. I am still a little bit puzzle by the difference of 27% between the size estimation of the genome and the final size. The author now highlight the difference and answer, but it is still something I am puzzled with.

I did not see LAI score and QVscore for assessing assembly quality, these statistics become more and more common and will be a nice add.

The author add nice additional analysis of population structure by chromosome to better support some of their claims about role of wild chromosomal fragment introgression.

We thank the reviewers for their comments. Our responses are provided below each comment. Please note that the line numbers refer to the document with track changes.

Reviewer #4 (Remarks to the Author):

Devos et al. report a chromosome scale genome assembly for the allotetraploid cereal finger millet and conducted downstream population and quantitative genetics related to domestication traits. I did not review the original version of this manuscript, but I have read this updated draft and the authors comments to the previous reviewers. I read this paper with interest, and think it represents a useful resource to help accelerate breeding of this important but underutilized cereal. Although the manuscript is generally improved, I think there are still a few issues that need to be addressed before publication.

Mainly, I do not think the evidence for identifying the gene underlying purple stigma to be very strong, and on a deeper level, I do not understand the significance of this trait to finger millet breeding or domestication. In line 313 the authors claim this phenotype is used for breeding, but they do not provide a source for this.

As part of this project, we worked extensively with finger millet breeders in eastern Africa (they are included as co-authors on the manuscript) and it was them who informed us of the importance of morphological markers, including purple coloration, in the identification of F_1 hybrids. We previously stated on lines 447-448 '*...in anthocyanin levels observed in stigma and anthers in finger millet germplasm, a widely used visual marker during breeding*'. We are now more specific in our statement '*...in anthocyanin levels observed in stigma and anthers in finger millet germplasm, a visual marker widely used by breeders to identify F_1 hybrids.*' We also changed the wording on lines 297-298 from '*Purple coloration of internodes, stigma and anthers has traditionally been used by breeders*' To '*Purple coloration of internodes, stigma and anthers is widely used by breeders*'. We don't have a reference but the statement is based on testimony of several of the manuscript's co-authors who are active finger millet breeders.

Stigma color is also independent to purple/brown seeds, so I am not sure what the agronomic relevance is for this. Perhaps this is a misunderstanding on my part, but readers will also probably not be experts in finger millet breeding, so a clearer connection to this trait and improvement is needed.

We wanted to examine whether finger millet accessions that lacked anthocyanins could still produce proanthocyanidins. As stated on line 497, the high proanthocyanidin levels in finger millet are considered to have health benefits. We moved into the realm of seed color (which does not necessarily equate to proanthocyanidin levels) when our analyses, geared at demonstrating that lines with a full-length *PP* gene could still lack anthocyanins if a mutation occurred in a biosynthetic gene, showed that knockout of two homoeologous chalcone synthase genes likely explained the lack of anthocyanins and likely also resulted in white seed. However, while most brown seeds have high proanthocyanidin levels, some brown-seeded lines have, as referenced on lines 499-500, low proanthocyanidin levels. White seeds have also been reported to have low proanthocyanidin levels, and while this is likely the case when white seed color is caused by knock-out of the chalcone synthase (*CHS*) gene, this has not been investigated for white lines that are caused by mutations in genes other than *CHS*. The situation is clearly complex, not fully resolved and is a separate story. To clarify why we looked at proanthocyanidin

levels, we added the following sentence (lines 375-376): *“This has implications for breeding for seed quality because proanthocyanidins have been associated with multiple health benefits⁴¹”*.

Importantly, the authors also state that only 15 out of 22 accessions with the functional PP allele displayed purple coloration and hypothesize that the other seven accessions have other non-functional alleles or deleterious mutations in other genes of the anthocyanin biosynthesis pathway. They provide some evidence for this in their revision by sequencing one of the white lines and identifying two possible loss of function mutations in a chalcone synthase gene. This is an interesting hypothesis and set of analyses, but I disagree that this overwhelmingly supports the MYC-bHLH transcription factor to be the causal gene underlying the QTL that controls purple stigma, as stated by the authors. Including these analyses in the paper is certainly fine, but in my opinion, these results do not ‘reveal the genetic basis of purple stigma color’ as claimed in the title and abstract, and the paper should be reframed as such to reflect this uncertainty. The authors specifically state this trait evolved at least twice through loss of functions in MYC-bHLH, but no downstream functional validation was done to verify this. I understand the difficulties with transformation of this species, but the use of phrases such as ‘likely’, or ‘candidate’ would be sufficient to address this major concern.

Although we disagree with the reviewer that the multiple lines of evidence that we provide do not overwhelmingly support the MYC-bHLH transcription factor as the causal gene that controls purple stigma (two independent mutations in the same gene associated with the same phenotype in a mutagenesis project are generally accepted as proof of function; we have QTL data as well as independent function-inactivating mutations in homoeologous genes), we have added ‘*candidate gene*’ to the title and shortened the title (to remain within the 15-word limit. The title now reads ‘*Genome analyses reveal population structure and a purple stigma color gene candidate in finger millet*’. In the abstract, we changed ‘*Loss of purple coloration of anthers and stigma occurred through loss-of-function mutations ...*’ to ‘*Loss of purple coloration of anthers and stigma was associated with loss-of-function mutations ...*’. We also changed the text in line 99 from ‘... causal gene ...’ to ‘... causal gene candidate ...’, in lines 452-453 from ‘*The gene PP is a MYC-bHLH transcription factor ...*’ to ‘*Our data strongly suggest that PP is a MYC-bHLH transcription factor ...*’, in lines 460-461 from ‘...only a single bHLH transcription factor, PP, regulates anthocyanin production ...’ to ‘...only a single bHLH transcription factor, PP, is associated with anthocyanin production ...’ and in lines 467-468 from ‘... partial deletions ... leading to the loss’ to ‘... partial deletions that are associated with the loss’.

Minor comments:

Line 429. This is not true, and a number of other chloridoid species have been sequenced including several *Zoysia* sp. *Eragrostis curvula*, *Eragrostis nindensis*, and several *Sporobolus* species, among probably others. Although lower quality, other finger millet genomes are also available but are still not really mentioned here.

We changed the text on lines 401-402 from ‘...*third Chloridoid species to have its genome sequenced*’ to ‘...*third Chloridoid species to have a reference-quality chromosome-resolved genome assembly generated*’. The text on lines 406-407 was changed from ‘... *than that of the other Chloridoids sequenced ...*’ to ‘...*than that of the other Chloridoids with chromosome-level genome assemblies ...*’. On line 404, we state that the size of the genome assembly is in line with earlier published genome assemblies, and these are referenced. We also clearly state at the beginning of the paper (lines 105-

106) that two other short-read assemblies are available for finger millet and that these have proven useful for broad-scale analyses.

Line 435. I'm not sure this is accurate. Contig N50 is becoming a meaningless statistic by itself without factoring in other characteristics such as genome size or chromosome number. For instance, a larger genome with a lower chromosome number could have a higher N50 but be more fragmented than a smaller genome.

We disagree that genome contiguity is becoming a meaningless statistic. Perhaps the reviewer is thinking of Scaffold N50, which is the statistic that is driven purely by chromosome sizes in a chromosome-scale genome? The primary predictors of genome quality are the number of contigs-per-chromosome and per-base quality (see below). This said, we are not set on contigN50 as the statistic to show this, and we could certainly swap "N50" for "n. contigs / chromosome". However, the former is more well understood and accepted by the community, which is why we presented it in the first place.

Reviewer #5 (Remarks to the Author):

Review of the article Devos et al. genome analyses reveal the genetic basis of population structure and purple stigma color in finger millet. The article is highly relevant to agriculture application, adaptation of agriculture to climate and study of neglected crops.

The article present a new assembly of the finger millet genome with better continuity, an analysis of A and B chromosome and a population structure shape by pericentromerics region, finally some likely candidate for purple color variation in the flower.

Minor comments

Overall, the authors answer to the comment in a satisfying way. I am still a little bit puzzle by the difference of 27% between the size estimation of the genome and the final size. The author now highlight the difference and answer, but it is still something I am puzzled with.

I did not see LAI score and QVscore for assessing assembly quality, these statistics become more and more common and will be a nice add.

LTR Assembly Index (LAI) scores, a measure of the 'completeness' of LTR retrotransposons, were initially introduced to compare the quality of short-read assemblies to long-read assemblies, but have lost meaning now that all high-quality assemblies are long-read based and this measure is rarely if ever included in genome papers. However, we agree that the QV score is a useful quality measure and have added this in the main document on line 116-117. The description on how the QV score was calculated is provided as Supplementary Note 1.

The author add nice additional analysis of population structure by chromosome to better support some of their claims about role of wild chromosomal fragment introgression.

Thank you.

Reviewers' Comments:

Reviewer #4:

Remarks to the Author:

I thank the authors for their response to my comments, the paper is greatly improved. I am excited to see this paper published, it will be an excellent resource for the community.

Reviewer #5:

Remarks to the Author:

Dear author

Thanks for answering my questions

I recommend publication

Best